# A hPSC-based platform to discover gene-environment interactions that impact human β-cell and dopamine neuron survival

Ting Zhou[1], Tae Wan Kim[2,3], Chi Nok Chong[1], Lei Tan[1,4], Sadaf Amin[1], Zohreh Sadat Badieyan[1], Suranjit Mukherjee[1], Zaniar Ghazizadeh[1], Hui Zeng[1], Min Guo[1], Miguel Crespo[1], Tuo Zhang [5], Reyn Kenyon[1], Christopher L. Robinson[1], Effie Apostolou[6], Hui Wang[4], Jenny Zhaoying Xiang[5], Todd Evans[1], Lorenz Studer[2,3] & Shuibing Chen[1,7]

Common disorders, including diabetes and Parkinson's disease, are caused by a combination of environmental factors and genetic susceptibility. However, defining the mechanisms underlying gene-environment interactions has been challenging due to the lack of a suitable experimental platform. Using pancreatic β-like cells derived from human pluripotent stem cells (hPSCs), we discovered that a commonly used pesticide, propargite, induces pancreatic β-cell death, a pathological hallmark of diabetes. Screening a panel of diverse hPSC-derived cell types we extended this observation to a similar susceptibility in midbrain dopamine neurons, a cell type affected in Parkinson's disease. We assessed gene-environment interactions using isogenic hPSC lines for genetic variants associated with diabetes and Parkinson's disease. We found $GSTT1^{-/-}$ pancreatic β-like cells and dopamine neurons were both hypersensitive to propargite-induced cell death. Our study identifies an environmental chemical that contributes to human β-cell and dopamine neuron loss and validates a novel hPSC-based platform for determining gene-environment interactions.

[1] Department of Surgery, Weill Cornell Medical College, 1300 York Ave, New York 10065 NY, USA. [2] The Center for Stem Cell Biology, Sloan-Kettering Institute for Cancer Research, New York, NY 10065, USA. [3] Developmental Biology Program, Sloan-Kettering Institute for Cancer Research, New York, NY 10065, USA. [4] School of Public health, Shanghai Jiao Tong University, School of Medicine, Shanghai 200000, China. [5] Genomic Resource Core Facility, Weill Cornell Medical College, 1300 York Ave, New York 10065 NY, USA. [6] Department of Medicine, Weill Cornell Medical College, 1300 York Ave, New York 10065 NY, USA. [7] Department of Biochemistry, Weill Cornell Medical College, 1300 York Ave, New York 10065 NY, USA. These authors contributed equally: Ting Zhou, Tae Wan Kim. Correspondence and requests for materials should be addressed to L.S. (email: studerl@mskcc.org) or to S.C. (email: shc2034@med.cornell.edu)

Diabetes is a polygenetic disease affected by both genetic and environmental factors, with the pathological hallmark of pancreatic β-cell death or failure. Genome-wide association studies have identified more than 80 candidate gene variants associated with diabetes[1–5]. A large number of these diabetes-associated genes are expressed in pancreatic β-cells[6], which highlights the importance of pancreatic β-cells themselves in the pathogenesis of diabetes. However, <10% of genetically susceptible individuals progress to type 1 diabetes[7], and gene variants explain a very small proportion of type 2 diabetes risk[8], which emphasizes the contribution of environmental factors in triggering or accelerating pancreatic β-cell loss. A recent report on U.S. synthetic chemical production suggested a causative role for synthetic chemicals in the pathogenesis of diabetes[9]. Indeed, several epidemiological studies linked environmental chemical toxins, such as polychlorinated biphenyls[10,11], bisphenol A[12–15], and heavy metals including arsenic[16], to an increased prevalence of diabetes. However, no systematic investigations on the role of environmental chemicals in human pancreatic β-cell death have been reported, due to the lack of an appropriate, robust, time-efficient and cost-efficient screening platform. In addition, the response to environmental toxins may vary based on genetic background. Thus, how environmental factors interact with candidate genes and contribute to disease progression remains largely unknown.

Understanding gene–environment interactions is critical to decode disease progression and develop novel precision therapies in diabetes as well as in neurodegenerative disorders such as Parkinson's disease (PD). Most of the current gene–environment interactions are examined via human population studies, which are complicated by the diverse genetic backgrounds of the subjects and the myriad of environmental conditions to which those individuals are exposed. Such studies typically require extremely large sample sizes to identify the interaction of genetic and environmental factors. A recent study using isogenic induced pluripotent stem cells (iPSCs) from PD patients provided a preview for the potential of using human embryonic stem cells (hESCs) and iPSCs to study the gene–environment interactions in disease pathogenesis[17]. Here, we combined directed differentiation of hESCs with gene-editing techniques to derive isogenic human pancreatic β-like cells carrying diabetes-associated variants, and used these to study gene–environment interactions relevant to β-cell survival. Notably, we found that these interactions similarly apply to midbrain dopamine neurons in the context of Parkinson's disease. The results suggest previously unappreciated similarities in the susceptibility of pancreatic β-cells and midbrain dopamine neurons to certain environmental toxins, and indicate broad applicability of our hPSC-based platform.

## Results

**A HTS to identify chemicals that target human β-cells.** We first sought to systematically explore the effects of environmental chemicals on human pancreatic β-cell survival using hESC-derived insulin-expressing (INS+) β-like cells. To perform the chemical screen, H1 hESCs were differentiated toward INS+ cell fate following our previously reported stepwise differentiation protocol (see Methods, pancreatic β-cell differentiation protocol 1): generating first SOX17+/FOXA2+ definitive endoderm, followed by PDX1+/NKX6.1+ pancreatic progenitors and finally PDX1+/INS+ cells (Fig. 1a and Supplementary Fig. 1a)[18]. The differentiated cell population containing ~25% INS+ cells and ~75% INS− cells were dissociated and re-plated on laminin V-coated 384-well plates for the chemical screen. The goal was to identify compounds that target a relative loss of the INS+ cells.

The Phase I Toxicity Forecaster (ToxCast) library provided by the U.S. Environmental Protection Agency (EPA) was used, which represents ~2000 compounds, including pesticides, industrial and consumer products. After overnight incubation, the chemicals were added at 20 nM, 200 nM, 2 μM, and 20 μM (detailed screening protocol and library information is described in the Methods). After 96 h of treatment, the cells were stained with an insulin antibody and analyzed using an ImageXpress[MICRO] Automated High-Content Analysis System. The chemicals that caused more than 60% reduction in the survival rate of INS+ cells, while affecting <20% loss of the INS− cells were picked as primary hits (Fig. 1a and Supplementary Fig. 1b). Two hit compounds were confirmed, including a rodenticide called N-3 pyridylmethyl-N''-4-nitrophenyl urea (Vacor) (Fig. 1b) that had previously been shown to be toxic to human pancreatic β-cells[19], and a pesticide, 2-(4-tert-butylphenoxy)-cyclohexyl prop-2-yne-1-sulfonate (propargite) (Fig. 1b). The lower IC$_{50}$ of both propargite and Vacor for INS+ cells as compared to INS− cells confirmed that these hit compounds specifically decreased the survival of INS+ cells (propargite: IC$_{50}$ = 1.09 μM for INS+ cells; IC$_{50}$ = 9.83 μM for INS− cells; Vacor: IC$_{50}$ = 4.89 μM for INS+ cells; IC$_{50}$ = 61.56 μM for INS− cells, Fig. 1c). Since Vacor has been banned by the EPA since 1979, we focused on the still-commonly used pesticide, propargite.

Using a related protocol recently published by our group[20] (see Methods, pancreatic β-cell differentiation protocol 2), hESCs were differentiated to glucose-responsive, pancreatic β-like cells (Supplementary Fig. 1c, d) to further examine the toxicity of propargite. Consistent with the primary screening results, propargite specifically decreased the number of glucose-responsive, pancreatic β-like INS+ cells in a dose-dependent manner (Fig. 1d, e). To determine whether propargite functioned through the induction of cell death, by affecting cell proliferation, or by the inhibition of β-cell differentiation, live cell-imaging was applied to monitor in real-time the survival of INS-GFP+ cells derived from INS$^{GFP/W}$ HES3 hESCs, which is an INS-GFP hESC reporter line[21]. INS-GFP+ cells gradually disappeared during the 4 days of propargite treatment, while the population of DMSO-treated INS-GFP+ cells did not change significantly (Fig. 1f, g). To exclude the possibility that propargite decreased the number of INS+ cells by affecting cell proliferation, DMSO and propargite-treated hESC-derived INS+ cells were stained for Ki67, a cell proliferation marker. Propargite did not affect the percentage of Ki67+/INS+ cells (Supplementary Fig. 1e, f). Due to the low proliferation rate of INS+ cells, we further examined the effect of propargite on MIN6 cells, a mouse pancreatic β-cell line that displays robust proliferation. Propargite-induced MIN6 cell death in a dose-dependent manner with IC$_{50}$ = 1 μM (Supplementary Fig. 1g), consistent with the effect on hESC-derived INS+ cells. No significant difference was detected between DMSO and 1.6 μM propargite-treated MIN6 cells regarding the percentage of Ki67+/INS+ cells after 4 days of treatment (Supplementary Fig. 1h, i), further validating that propargite does not function through blocking cell proliferation. Finally, primary human islets were treated for 4 days with DMSO or propargite and subsequently stained with propidium iodide (PI) to detect dead cells. Consistent with the effect on hESC-derived INS+ cells, propargite decreased the number of INS+ cells (Fig. 1h, i), accompanied with an increased percentage of PI+/INS+ cells in INS+ cells of primary human islets (Fig. 1h, j).

**Propargite induces β-cell necrosis preceded by DNA damage.** RNA-seq was used to compare global transcript expression profiles between DMSO and propargite-treated INS-GFP+ cells derived from INS$^{GFP/W}$ HES3 cells. Gene ontology pathway

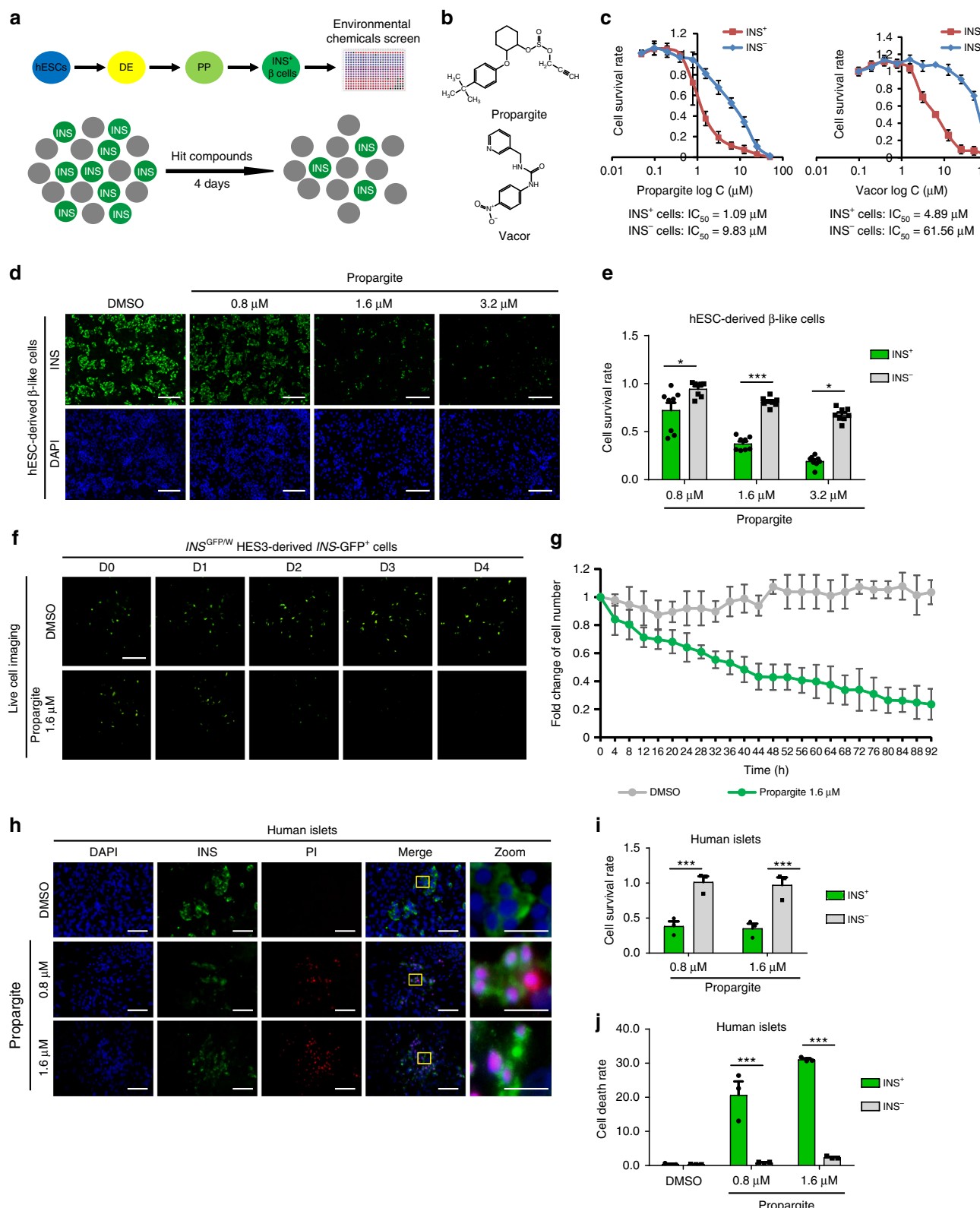

analysis showed "regulation of DNA-dependent transcription" among the top downregulated pathways upon Propargite treatment (Fig. 2a). In contrast, genes associated with chromatin assembly and cell death-related processes were upregulated under those conditions (Fig. 2a). Several genes that were highly upregulated (fold-change >3) in the propargite-treated cells were related to DNA damage, including, DNA damage-inducible

transcript 3 (*DDIT3*) and the growth arrest and DNA damage-inducible alpha (*GADD45A* or *DDIT1*) (Supplementary Fig. 2a and Table 1). The upregulation of these genes was further confirmed by qRT-PCR analysis (Fig. 2b). Propargite-treated MIN6 cells were next examined using a well-established DNA damage marker, phospho-histone H2A.X (γ-H2A.X)[22]. Western blotting confirmed that propargite-induced γ-H2A.X expression in a

**Fig. 1** A high-content screen identifies propargite as a compound that induces pancreatic β-cell death. **a** Scheme of pancreatic β-cell differentiation and high-content chemical screen. **b** Chemical structure of two hit compounds: Propargite and Vacor. **c** Inhibitory curve and $IC_{50}$ of propargite or Vacor on $INS^+$ and $INS^-$ cells ($n = 8$). **d, e** Immunocytochemistry analysis (**d**) and survival rate (**e**) of H1-derived glucose-responding $INS^+$ and $INS^-$ cells treated with different doses of propargite ($n = 8$). Pancreatic β-like cells were stained for INS (green) and all cells were counterstained by DAPI (blue). Scale bars, 100 μm. The survival rate was calculated by dividing the average number of cells in propargite-treated conditions by the average number of cells in the DMSO control. **f, g** Representative live images (**f**) and fold-change (**g**) of $INS^{w/GFP}$ HES3-derived $INS$-GFP$^+$ cells treated with 1.6 μM propargite or DMSO control at different time points compared to initial time 0 ($n = 3$). Scale bars, 100 μm. **h, i, j** Representative images (**h**) and cell survival rate (**i**) or cell death rate (**j**) of DMSO or propargite-treated human primary islets ($n = 3$). Pancreatic β-cells were stained with the INS antibody (green) and dead cells were stained with propidium iodide (PI) (red). Scale bars, 200 μm. Scale bars for higher resolution (Zoom) images, 800 μm. Pancreatic β-cell death rate was calculated by defining percentages of $INS^+$ $PI^+$ cells in $INS^+$ cells, and non-β-cell death rate was calculated by percentages of $INS^-PI^+$ cells in $INS^-$ cells. Values presented as mean ± S.D. n.s. indicates a non-significant difference. $p$ values calculated by unpaired two-tailed Student's $t$-test were *$p < 0.05$, ***$p < 0.001$. Related to Supplementary Fig. 1

dose-dependent and time-dependent manner (Fig. 2c), indicating that propargite induces DNA damage in pancreatic β-cells.

To define the mechanism of propargite-induced cell death, propargite-treated β-like cells were examined for apoptosis, necrosis, and autophagy[23]. First, flow cytometry was used to examine phosphatidylserine (PS) translocation with anti-Annexin-V antibodies, and immunocytochemistry was used to determine caspase-3 activation. Propargite induced neither PS translocation (Supplementary Fig. 2b, c) nor caspase-3 activation (Supplementary Fig. 2d, e), which indicates that apoptosis is not the primary cause of propargite-induced β-cell death. Second, in western blotting experiments we found no significant increase in the relative levels of LC3-II, a widely used marker of autophagy, in propargite-treated MIN6 cells, suggesting that propargite does not function through inducing autophagy (Supplementary Fig. 2f)[24]. Finally, we measured extracellular amounts of high mobility group B1 (HMGB1) protein, which is chromatin-bound under normal circumstances and released into the extracellular milieu when a cell undergoes necrosis[25,26]. Western blotting experiments detected extracellular HMGB1 as early as 1 day after propargite treatment, and the levels increased progressively over time. Additionally, the levels of extracellular HMGB1 were elevated in a dose–response manner to propargite while intracellular HMGB1 levels decreased or remained constant (Fig. 2d). To further validate necrosis as the cause of cell death, a time course experiment was performed, revealing that γ-H2A.X was induced 12 h after treatment of MIN6 cells, before the detection of extracellular HMGB1 at 18 h after treatment (Fig. 2e). Thus, propargite initiates DNA damage on β-cells and subsequently causes cell death by necrosis.

**β-cells are sensitive to propargite due to low glutathione**. A previous study found that in rats propargite and propargite-immediate metabolites directly conjugated with glutathione (GSH) before glutathione S-transferases (GST) enzyme detoxification and degradation (http://www.fao.org/fileadmin/templates/agphome/documents/Pests_Pesticides/JMPR/Evaluation02/propargiteevaljj.pdf). To determine whether the hypersensitivity of $INS^+$ pancreatic β-cells was due to low glutathione (GSH) levels, the endogenous GSH levels were measured in hESC-derived $INS$-GFP$^+$ and $INS^-$ cells, primary human islets, and BJ-fibroblasts. The $INS$-GFP$^+$ cells and human islets had significantly lower GSH levels compared to those of $INS^-$ and fibroblast cells (Fig. 2f). To determine whether low GSH levels caused hypersensitivity to propargite treatment, GSH was added to cultured cells in an attempt to rescue propargite-induced β-cell toxicity. Indeed, 2 mM GSH fully rescued the survival rate of $INS^+$ cells after propargite treatment (Fig. 2g, h). In addition, GSH also rescued the propargite-induced pancreatic β-cell-specific death in human islets (Fig. 2i, j). Furthermore, RNA-seq profiles showed that GSH reversed propargite-induced global

transcriptional changes (Fig. 2k, l), including the down-regulation of genes related to pancreas function and the upregulation of genes associated with cell death (Supplementary Fig. 3a). Additionally, GSH fully rescued propargite-induced DNA damage (indicated by γ-H2A.X expression levels) and partially rescued propargite-induced necrosis, as indicated by extracellular HMGB1 protein levels (Fig. 2m). Since GSH is an important anti-oxidant, the propargite-treated MIN6 cells were monitored for reactive oxygen species (ROS) (Supplementary Fig. 3b). However, we did not find any increase of ROS in propargite-treated conditions, suggesting the effect of GSH on rescuing β-like cells from propargite-induced necrosis was not due to its antioxidative properties. Previous findings in rodents reported on a GSH-dependent propargite detoxification pathway (Supplementary Fig. 3c), suggesting that a similar mechanism may apply to its action in pancreatic β-cells.

**$GSTT1^{-/-}$ β-cells show increased susceptibility to propargite**. We next sought to test whether the genetic background of β-like cells affects their response to propargite-induced toxicity, with a particular interest in whether diabetes-associated gene variants could impact their sensitivity. Based on the proposed mode-of-action of propargite, the genes encoding GSTs, a family of phase II metabolic isozymes, known for the ability to catalyze the conjugation GSH to xenobiotic substrates (Supplementary Fig. 3c)[27], is a strong candidate to consider. Among more than six different types of cytosolic GSTs[27], the polymorphic homozygous deletions of GST mu1 (GSTM1) and GST theta1 (GSTT1) occurs with a frequency of ~40–60% and ~13–25%, respectively, in Caucasian populations[28]. GWAS data suggest that this GSTT1-null and GSTM1-null genotype are associated with increased risk of diabetes[29–31]. Therefore, we performed a small population study using pancreatic β-like cells derived from three hESCs and seven hiPSCs. Indeed, hESC/iPSC derived pancreatic β-like cells displayed variable survival rates in the presence of 1.6 μM propargite treatment (Fig. 3a). Genotyping of the GSTM1 and GSTT1 genes by PCR showed that of the ten cell lines four are null for both GSTM1 and GSTT1, and 2 are null for GSTM1 only, consistent with the heterogeneity of GSTM1 and GSTT1 expression in the human genome (Fig. 3a). A correlation study was performed to compare the survival rates of $INS^+$ cells derived from these lines following propargite treatment. The $INS^+$ cells from those lines lacking both genes have significantly lower survival rates compared to cells derived from those lines having at least the GSTT1 gene present (Fig. 3b), while GSTM1-null pancreatic β-like cells did not show significantly lower survival rates compare to lines with an intact GSTM1 gene (Fig. 3c). To further validate these results in an isogeneic background, $GSTM1^{-/-}$ and $GSTT1^{-/-}$ hESCs were established using a CRIPSR-based genome editing approach in the H1 parental cell background. Considering potential variation between different

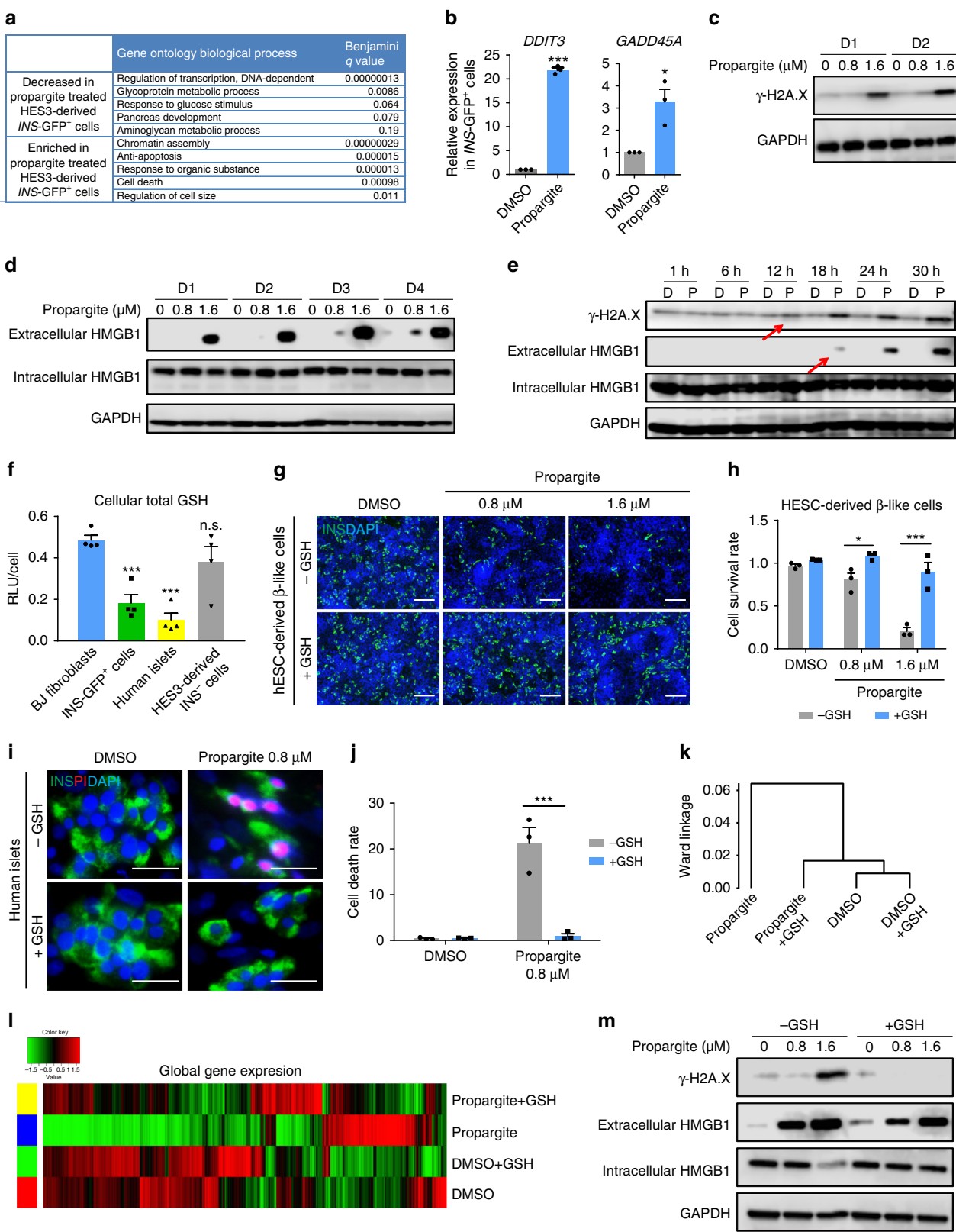

hESC clones, two clones from each line were chosen for further analysis. Biallelic indel mutations for each of the targeted genes were validated by genomic DNA sequencing (Supplementary Fig. 3d, e). The two GSTT1 knockout clones were both homozygous null mutants, and the two GSTM1 knockout clones were both compound-null mutants. Each indel mutation created an early frameshift that generated null alleles as confirmed by western blotting experiments in mutant hESC-derived cells (Fig. 3d).

The wild type (WT), $GSTM1^{-/-}$, and $GSTT1^{-/-}$ hESCs had similar capacities to differentiate into INS$^+$ cells as monitored by intracellular FACS (Fig. 3e), suggesting that the $GSTM1^{-/-}$ and $GSTT1^{-/-}$ mutations did not affect differentiation to INS$^+$ cells.

**Fig. 2** Propargite induces glutathione (GSH)-dependent β-cell necrosis preceded by DNA damage. **a** Top five significantly upregulated and downregulated biological processes identified by RNA-seq profiles in propargite-treated $INS^{w/GFP}$ HES3-derived $INS$-GFP+ cells. **b** qRT-PCR analysis of $DDIT3$ and $GADD45A$ in DMSO or propargite-treated $INS$-GFP+ cells ($n = 3$). **c** Western blotting analysis of γ-H2A.X in MIN6 cells treated with propargite. **d** Western blotting analysis of necrosis marker (extracellular HMGB1) in MIN6 cells treated with propargite. **e** Time course western blotting analysis of extracellular HMGB1 and γ-H2A.X of MIN6 cells treated with DMSO (D) or propargite (P) during a period of 1 to 30 h. GAPDH was used as a loading control in all the western blots. Red arrows indicate initiation of propargite-induced changes. **f** Total GSH levels in human fibroblasts, $INS^{w/GFP}$ HES3-derived $INS$+ cells, $INS$− cells and human islets ($n = 4$). **g**, **h** Representative images (**g**) and cell survival rate (**h**) of DMSO or propargite-treated H1-derived $INS$+ cells in the presence of absence of 2 mM GSH ($n = 3$). Scale bars, 100 μm. **i**, **j** Representative images (**i**) and cell death rate (**j**) of DMSO or propargite-treated human islets in the presence or absence of 2 mM GSH ($n = 3$). Scale bars, 800 μm. **k**, **l** Hierarchical clustering (**k**) and heatmap (**l**) of transcriptional profiles in $INS^{w/GFP}$ HES3-derived $INS$-GFP+ cells treated with DMSO, DMSO+ 2 mM GSH, 1.6 μM propargite, or 1.6 μM propargite+2 mM GSH. **m** Western blotting analysis of extracellular HMGB1 and γ-H2A.X in propargite-treated MIN6 cells on day 4. Values presented as mean ± S.D. n.s. indicates a non-significant difference. $p$ values calculated by unpaired two-tailed Student's $t$-test were *$p < 0.05$, ***$p < 0.001$. Related to Supplementary Figs. 2 and 3

To generate propargite inhibitory curves, cells derived from the WT, $GSTM1^{−/−}$ or $GSTT1^{−/−}$ hESCs were treated with propargite at increasing concentrations. The $IC_{50}$ value of propargite for the $GSTT1^{−/−}$ $INS$+ cells (but not the $GSTM1^{−/−}$ $INS$+ cells) was significantly lower than that of the WT $INS$+ cells ($IC_{50} = 1.17$ μM for $GSTT1^{−/−}$-1 $INS$+ cells; $IC_{50} = 2.15$ μM of $GSTT1^{+/+}$-1 $INS$+ cells) (Fig. 3f). Meanwhile, lower survival rates were accompanied by significantly higher levels of DNA damage in the cells as determined by the percentages of γ-H2AX+ $INS$+ /$INS$+ cells (Fig. 3g, h). Together, these results indicate that $GSTT1$ gene deletion increases the susceptibility of $INS$+ cells to propargite-induced cell death. To further confirm our results in mature human pancreatic β-cells, we used a human β-cell line, EndoC-βH1 cells[32], that was confirmed positive for the GSTT1 gene (Supplementary Fig. 3f). The cells were infected with a lentivirus expressing the Cas9 gene and two sgRNAs targeting different locations of exon 4 in the $GSTT1$ gene (Supplementary Table 2 and 3). Loss of GSTT1 protein expression in EndoC-βH1 cells carrying sg$GSTT1$ was confirmed by western blotting experiments (Fig. 3i). EndoC-βH1 cells carrying sg$GSTT1$ were subsequently cultured in the presence or absence of propargite and analyzed with PI staining. Consistent with $GSTT1^{−/−}$ hESC-derived $INS$+ cells, a significant increase of $INS$+ cell death was detected in EndoC-βH1 cells expressing sg$GSTT1$ (Fig. 3j, k). Accordingly, our results point to the fact that the toxicity of propargite is strongly modulated by the host cell genetic background.

**Midbrain dopamine neurons are hypersensitive to propargite.** Next, we determined whether propargite is toxic to other cell types by treating a range of different hESC-derived lineages with 1.6 μM of propargite, including CD29+/CD73+ mesenchymal stem cells, CTNT+ cardiomyocytes, A1AT+ hepatocytes, HuC/D+ neurons and primary BJ-fibroblasts (Supplementary Fig. 4a–e). While most cell types exhibited a low sensitivity to propargite, we found that HuC/D+ neurons were highly susceptible to propargite similar to the results obtained in β-cells (Supplementary Fig. 4f–h). This raised the question whether propargite exposure could be associated with neurodegenerative disease.

An epidemiological study has reported that high levels of well-water contamination with propargite is associated with an increased prevalence of Parkinson's disease in the affected region[33]. Parkinson's disease is characterized by the specific loss of midbrain dopamine (mDA) neurons[34,35]. Therefore, we hypothesized that mDA neurons may be particularly sensitive to propargite toxicity. To test this hypothesis, we derived TH+/FOXA2+ mDA neurons and MAP2+/CTIP2+ cortical neurons from hESCs using differentiation conditions published previously by our group[36,37] (Fig. 4a and Supplementary Fig. 5a–c). Cell

survival data showed that mDA neurons were dramatically more sensitive to propargite than cortical neurons resulting in significant mDA neuron cell death when treated with concentrations at 1 and 3 μM (Fig. 4b). We next examined whether propargite-induced toxicity in mDA neurons acts via a mechanism similar to that observed for β-cell death. Indeed, GSH was able to rescue mDA neuron death in response to propargite treatment (Fig. 4c, d) by preventing both DNA damage and cell necrosis (Fig. 4e, f). Additionally, $GSTT1^{−/−}$ hESC-derived mDA neurons, assessed by the expression of FOXA2, were more sensitive to propargite than matched FOXA2+ neurons derived from isogenic $GSTT1^{+/+}$ or $GSTM1^{−/−}$ hESCs, consistent with the data using β-cells (Fig. 4g and Supplementary Fig. 5d). Analysis of published gene expression data from substantia nigra tissue of PD patients[38] shows a decrease in expression levels for $GSTT1$ but not $GSTM1$, compared to age-matched controls (Fig. 4h). This suggests that a decreased GSTT1 level is a potential risk factor for PD, mediated by responses to environmental stimuli. Collectively, our results indicate that propargite exerts very similar effects in both β-cells and mDA neurons, and the findings provide a potential mechanism for how increased propargite levels in the drinking water may increase the prevalence of diabetes or PD[39].

**$GSTT1^{−/−}$ β-cells are hypersensitive to propargite in vivo.** To investigate whether β-cell toxicity is caused by propargite in vivo, CD-1 mice were treated with different doses of propargite or corn oil (vehicle) daily for 5 days via intraperitoneal injection. 12 mg/kg propargite treatment lightly increased blood glucose level without significantly affecting body weight (Supplementary Fig. 6a, b). Meanwhile, it led to a significant reduction of insulin levels in the plasma of the fasted mice (Fig. 5a). A significant increase of the percentage of γ-H2AX+/$INS$+ in $INS$+ cells of propargite-treated mice was also observed, suggesting enhanced rates of propargite-induced DNA damage in mouse pancreatic β-cells in vivo (Fig. 5b, c). To investigate the effect of propargite on human pancreatic β-cells in vivo, we infected EndoC-βH1 cells with a lentivirus carrying constitutively expressed luciferase to generate the luciferase-expressing EndoC-βH1 cells (EndoC-βH1-Luc cells). The EndoC-βH1-Luc cells were then transplanted into NSG immune-deficient mice to create a humanized mouse model (Fig. 5d). The humanized mice were administrated with 12 mg/kg propargite. In vivo imaging for luciferase signal showed that luciferase signals from control EndoC-βH1-Luc cells increased gradually during the 5-day course. In contrast, the signal of cells in the propargite administered mice showed a slower rate increase at day 3 and a marked decrease at day 5 (Fig. 5e and Supplementary Fig. 6c). To further confirm that the sensitivity of β-cells to propargite depends on genetic background, EndoC-βH1-Luc cells carrying sg$GSTT1$ were transplanted into the immune-

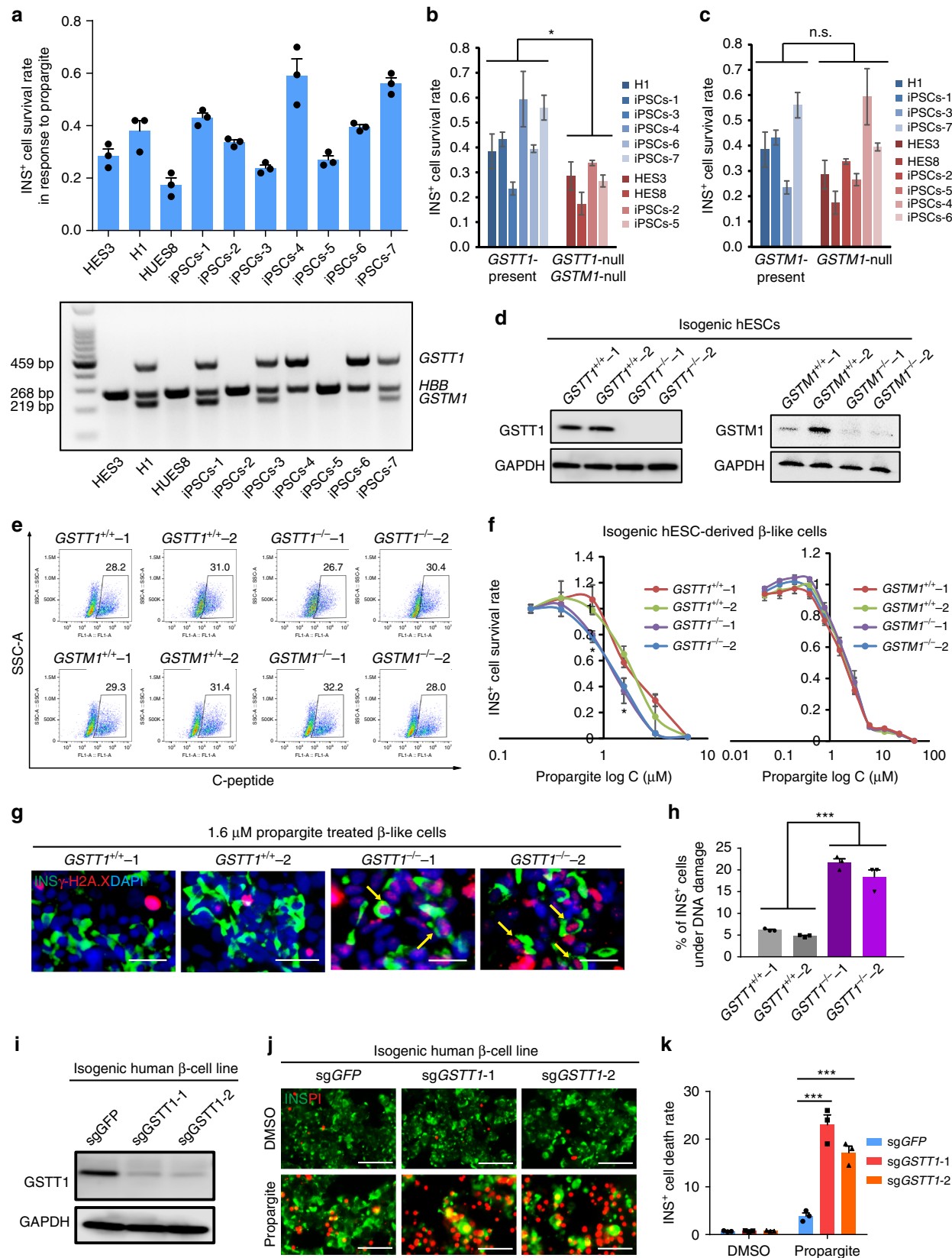

deficient mice. EndoC-βH1-Luc cells carrying sg*GFP* were used as a control. At day 3, the survival rate of the EndoC-βH1-Luc cells carrying sg*GSTT1* was associated with higher cell death rate upon propargite treatment as measured with TUNEL assays (Fig. 5f, g). Together, the data indicate that *GSTT1*$^{-/-}$ pancreatic β-like cells are hypersensitive to propargite in vivo.

## Discussion

GWAS have identified an increasing number of diabetes-associated genes, yet only a small proportion of individuals harboring these genomic variables progress to clinical disease. This implies that additional factors such as environmental influences, may drive disease initiation and progression in genetically

**Fig. 3** A hPSC-based population study discovers that *GSTT1*-null pancreatic β-like cells are hypersensitive to propargite-induced cell death. **a** Survival rate of INS$^+$ cells derived from 10 different hESC or iPSC lines cultured in the presence of 1.6 μM propargite (n = 3), and genotype analysis of *GSTM1* and *GSTT1* in those hESCs and iPSCs. **b, c** Correlation of INS$^+$ cell survival rate in the presence of 1.6 μM propargite in cells lacking both *GSTM1* and *GSTT1* (**b**), or lacking only *GSTM1* (**c**). n.s. indicates a non-significant difference. **d** Western blotting analysis of GSTT1 or GSTM1 protein expression in INS$^+$ cells derived from isogenic wild type, *GSTT1*$^{-/-}$ or *GSTM1*$^{-/-}$ H1 hESCs. The $-/-$ null clones were CRSIPR-induced biallelic frameshift mutants. The two GSTT1 knockout clones were both homozygous null mutants, and the two GSTM1 knockout clones were both compound-null mutants. **e** Flow cytometry analysis of C-peptide$^+$ cells in isogenic *GSTT1*$^{-/-}$ or *GSTM1*$^{-/-}$ hESC-derived D18 cells. **f** Inhibition curve of propargite on INS$^+$ cells derived from *GSTT1*$^{+/+}$ or *GSTT1*$^{-/-}$ H1 hESCs (n = 3). **g, h** Representative images (**g**) and DNA damage rate (**h**) of *GSTT1*$^{+/+}$ and *GSTT1*$^{-/-}$ β-like cells (n = 3). Scale bars, 800 μm. γ-H2A.X $^+$/INS$^+$ cells are highlighted with yellow arrows. **i** Western blot analysis of GSTT1 protein in EndoC-βH1 cells carrying *sgGSTT1*. Two CRISPR gRNAs (*sgGSTT1*-1 and *sgGSTT1*-2) were used for generating *GSTT1*$^{-/-}$ EndoC-βH1 cells. **j, k** Representative images (**j**) and cell death rate (**k**) of *GSTT1*$^{-/-}$ EndoC-βH1 cells treated with 1.6 μM propargite (n = 3). Scale bars, 200 μm. Values presented as mean ± S.D. n.s. indicates a non-significant difference. p values calculated by unpaired two-tailed Student's t-test were *p < 0.05, ***p < 0.001. Related to Supplementary Fig. 3

predisposed subjects. To model this scenario, we performed an unbiased high-content chemical screen and identified propargite, which induced pancreatic β-cell-specific death. The ubiquity of propargite as a common pesticide in pediatric patients is made clear by an NIH report stating that the average daily intake of propargite is 0.28 μg/kg in 6–11-month-old infants and 0.22 μg/kg in 2-year-old toddlers, which is around 5.5–7 times higher than the average daily intake in adults (http://toxnet.nlm.nih.gov/cgi-bin/sis/search/a?dbs+hsdb:@term+@DOCNO+1528). It has been reported that the 8-hour acute absorbed daily dosage (ADD) calculated for aerial applicators handling the wettable powder in water soluble bags was 5300 μg/day per kilogram (kg) of body weight (http://www.cdpr.ca.gov/docs/whs/pdf/hs1527.pdf). We used both mouse and humanized mouse models show that daily exposure of propargite causes the decrease of cell number and increase of DNA damage of both mouse and human β-cells. It worth to note that that gating strategy in the primary screening (60% reduction in the survival rate of INS$^+$ cells and <20% loss of the INS$^-$ cells) is very strict. In this study, we only focus on the environmental chemicals showing strong β-cell toxicity. The compounds showing weak acute β-cell toxicity might be missing. Further, we monitored the propargite's toxicity on different types of cells using hPSC-derived population. We suggested that propargite decreases cell number by affecting cell survival since most of the hPSC-derived population show relative high purity. We cannot fully exclude the possibility that propargite also affects hPSC differentiation.

Diabetes is a polygenic disease involving genetic and environmental factors. Although several epidemiology studies have reported gene–environment interactions in diabetes[40,41], human population-based studies are also complicated by heterogeneous genetic backgrounds and dynamic environmental conditions, requiring very large sample sizes to identify interaction of genetic and environmental factors. Here, we report the first hESC/iPSC-based population study to determine gene–environment interactions on pancreatic β-cell survival and found that *GSTT1*-null cells exhibit increased sensitivity to propargite-induced pancreatic β-cell death, which was further validated using isogenic *GSTT1*$^{-/-}$ hESC-derived cells and an isogenic human pancreatic β-cell line. Although both GSTT1-null and GSTM1-null genotype are associated with increased risk of diabetes[29–31] in GWAS, only *GSTT1*$^{-/-}$ β-like cells show increased sensitivity to propargite-induced cell death. One possible reason for this finding is that the expression levels of *GSTT1* and *GSTM1* might be different in human versus mouse islets. Notably, there was still a minimal signal in the compound-null GSTM1 knockout cells in the western-blot lanes. It could be due to the reduced stability of the indel-frameshift GSTM1 proteins, or the background of the GSTM1 antibody. Additional experiments will be required to more fully examine the differences between GSTT1-null and GSTM1-null cells. The hESC/iPSC-based system, in which both

pancreatic β-cell derivation and environmental factors can be tightly controlled, provides a robust and relatively high-throughput platform to systematically examine the interactions of diabetes risk-associated genes and environmental factors.

The fact that the same risk factors and mechanism of propargite-mediated cell death apply to hPSC-derived midbrain DA neurons indicates the broader applicability of our findings and point to a shared susceptibility profile of β-cells and mDA neurons. Supporting epidemiological data on PD prevalence in populations exposed to well water with high propargite levels links the in vitro findings to actual patient data[39]. Our hPSC-based in vitro platform can be used to probe the impact of environmental factors for public health, and may facilitate the development of novel precision therapies for patients at risk of developing diabetes or PD.

## Methods
**Cell culture**. hESCs were grown on Matrigel-coated plates with mTeSR1 medium (Stem Cell Technology). Cells were maintained at 37 °C with 5% $CO_2$. MIN6 cells were a gift from Dr. Mingming Hao. Human β-cell line (EndoC-βH1) was purchased from Endocells. H1 hESCs were purchased from WiCell Institute. HUES8 hESCs were provided by Harvard University. HES3 hESCs were provided by Dr. Ed Stanley at Monash University, Australia.

**hESC differentiation**. The starting population of cells for the high-content chemical screen were derived from H1 hESCs following our previously reported stepwise differentiation protocol (β-cell differentiation protocol 1)[18]. Hit compounds were validated using the cells derived using a strategy recently published by our lab[20] (β-cell differentiation protocol 2). Cortical and mDA differentiation from hESC were done with similar protocols those published by our lab previously[36,37].

**High-content screen**. The screening library ToxCast Phase I was obtained from the US Environmental Protection Agency, which contains approximately 2000 compounds dissolved at 20 mM in DMSO. The H1 hESC-derived population was dissociated with Accutase and re-plated onto laminin V-coated 384-well plates at 5000 cells per well. After 16 h incubation, cells were treated with individual compounds from the ToxCast library with four concentrations, including 20 nM, 200 nM, 2 μM, and 20 μM, for each compound. After 4 days, the cells were fixed with 10% formalin and stained with an INS antibody and DAPI, followed by automatic imaging and analysis by ImageXpress$^{MICRO}$ Automated High-Content Analysis System (Molecular Devices) to count the numbers of INS$^+$ and INS$^-$ cells. The survival rate was calculated by dividing the numbers of INS$^+$ cells in compound treated conditions by the average number of INS$^+$ cells in DMSO-treated conditions. The compounds that induced more than 60% reduction of the survival rate of INS$^+$ cells and less than 20% of the INS$^-$ cells were chosen as primary hits. Hit compounds were then selected and validated in three additional independent experiments using the same protocol as the primary screening.

**Generation of inhibitory curves**. hESCs-derived INS$^+$ cells[42], MSCs[43], neurons[37], and hepatocytes[44] were derived based on the reported protocols and re-plated on 96-well plates with Accutase digestion. Cardiomyocytes (CMs) were differentiated from H9-MYH6: mCherry reporter line, based on the reported protocol[45]. After overnight incubation, cells were treated with DMSO or different doses of hit compounds for 4 days. After fixation, the cells were stained with markers for each cell type. DAPI was used to stain cell nuclei. Plates were analyzed by ImageXpress$^{MICRO}$ Automated High-Content Analysis System.

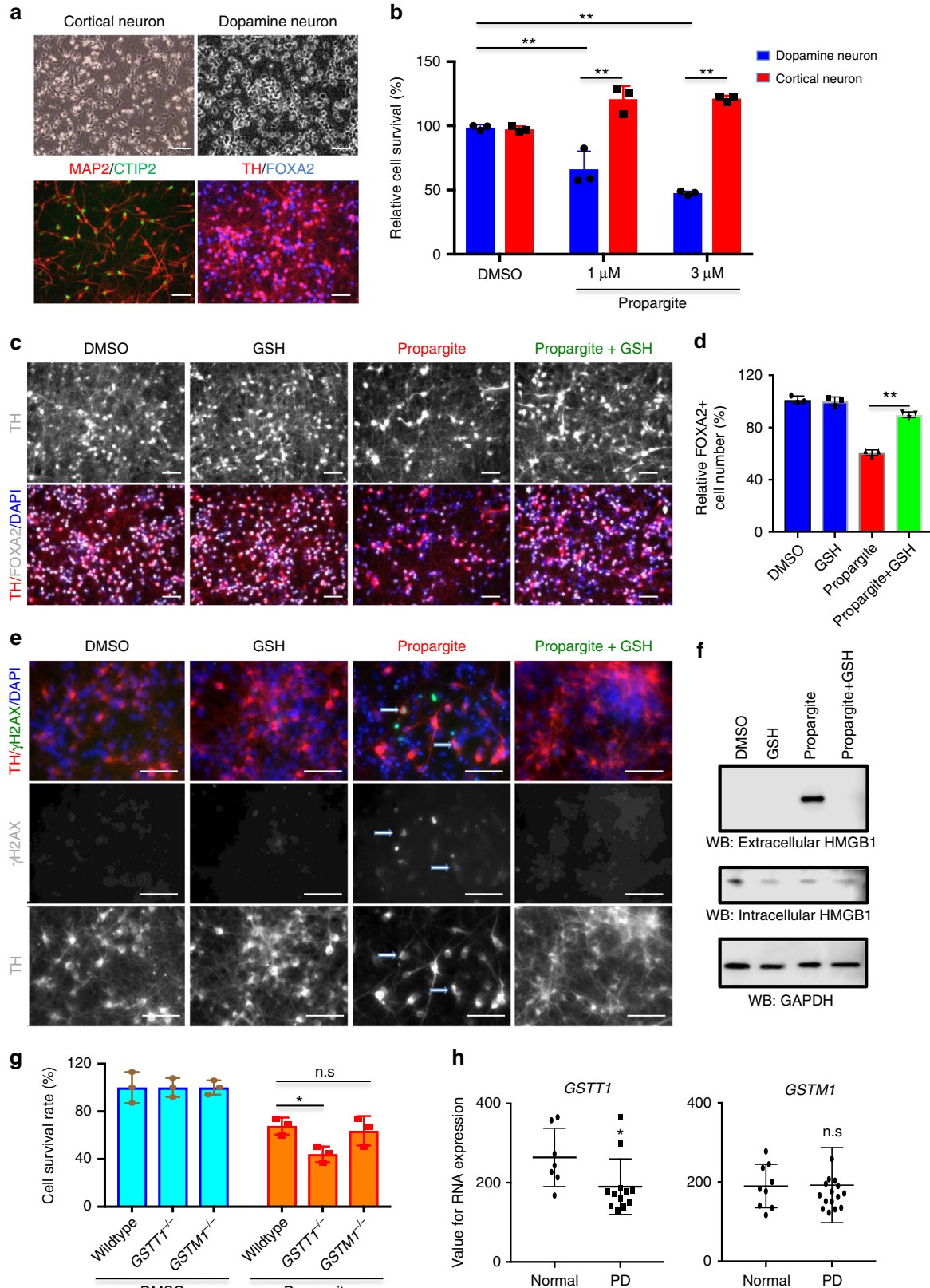

**Creation of isogenic hESC lines**. CRISPR sgRNA sequences were designed using the web resources available at http://www.genome-engineering.org/. The target sequences are listed in Supplementary Table 2. Each target sequence was cloned into the pX330-U6-Chimeric_BB-CBh-hSpCas9 vector (Addgene plasmid #42230) to make the gene targeting constructs. To knockout the target genes, two sgRNAs were validated using the surveyor assay in 293T cells, and we chose 1 sgRNA-construct plasmid for further experiments. After validation, H1 or $INS^{GFP/W}$ HES3

cells were dissociated using Accutase (STEM CELL) and electroporated ($8 \times 10^5$ cells per sample) with 4 μg sgRNA-construct plasmids using Human Stem Cell Nucleofector$^{TM}$ solution (Lonza) following manufacturer's instructions. The cells were then seeded into 2 wells of 24-well plates. 4 days later, hESCs were dissociated into single cells by Accutase (STEM CELL) and re-plated at a low density (1–3 cells per well in 96-well plates). 10 μM Y-27632 was added. 10 days later, individual colonies were picked, mechanically disaggregated and re-plated into two individual

**Fig. 4** Midbrain dopamine neurons are hypersensitive to propargite-induced cell toxicity. **a** Characterization of cortical neuron and mDA neuron derived from H9 hESCs. Upper panel represents bright field images of cortical- and mDA-neurons. Lower panel shows cortical neurons stained for MAP2 (red) and CTIP2 (green) while mDA neurons were stained for TH (red) and FOXA2 (blue). Scale bars, 50 μm. **b** Relative cell survival rate of cortical- and mDA-neurons treated with DMSO or different doses of propargite. Relative cell survival was quantified by dividing propargite-treated cells to the DMSO control ($n = 3$). **c**, **d** Representative image (**c**) and relative cell survival rate (**d**) of mDA neurons treated with DMSO or propargite (1 μM) in the presence or absence of GSH (2 mM). mDA cells were stained for TH (red) and FOXA2 (gray), and all cells were counterstained with DAPI (blue). Scale bars, 50 μm. Relative cell survival rate was analyzed by quantification of FOXA2$^+$ (gray) cells ($n = 3$). **e** Representative image of mDA cells treated with DMSO, DMSO +2 mM GSH, 1 μM propargite, or 1 μM propargite+2 mM GSH. White arrows indicate propargite-treated mDA cells (TH; red) co-stained with the DNA damage marker (rH2AX; green), and all cells were counterstained with DAPI (blue). Scale bars, 50 μm. **f** Western blotting analysis of necrosis marker (extracellular HMGB1) in DMSO or propargite (1 μM) treated mDA cells with/without GSH (2 mM). Only propargite-treated mDA cell had high extracellular HMGB1 level ($n = 3$). **g** Relative cell survival rate, quantified by the expression of FOXA2+ cells ($n = 3$), of mDA cells derived from isogenic wild type, $GSTT1^{-/-}$, and $GSTM1^{-/-}$ H1 hESCs treated with DMSO and propargite (3 μM). **h** $GSTT1$, but not $GSTM1$ expression, in substantia nigra region of postmortem brains is significantly downregulated in Parkinson's disease patients compared to age-matched controls. Values for RNA expression used here are from a published gene expression data and selected values except absent its detection[38]. Values presented as mean ± S.D. $p$-value was calculated by unpaired two-tailed Student's $t$-test were *$p < 0.05$, **$p < 0.01$. n.s. indicates a non-significant difference. Related to Supplementary Figs. 4 and 5

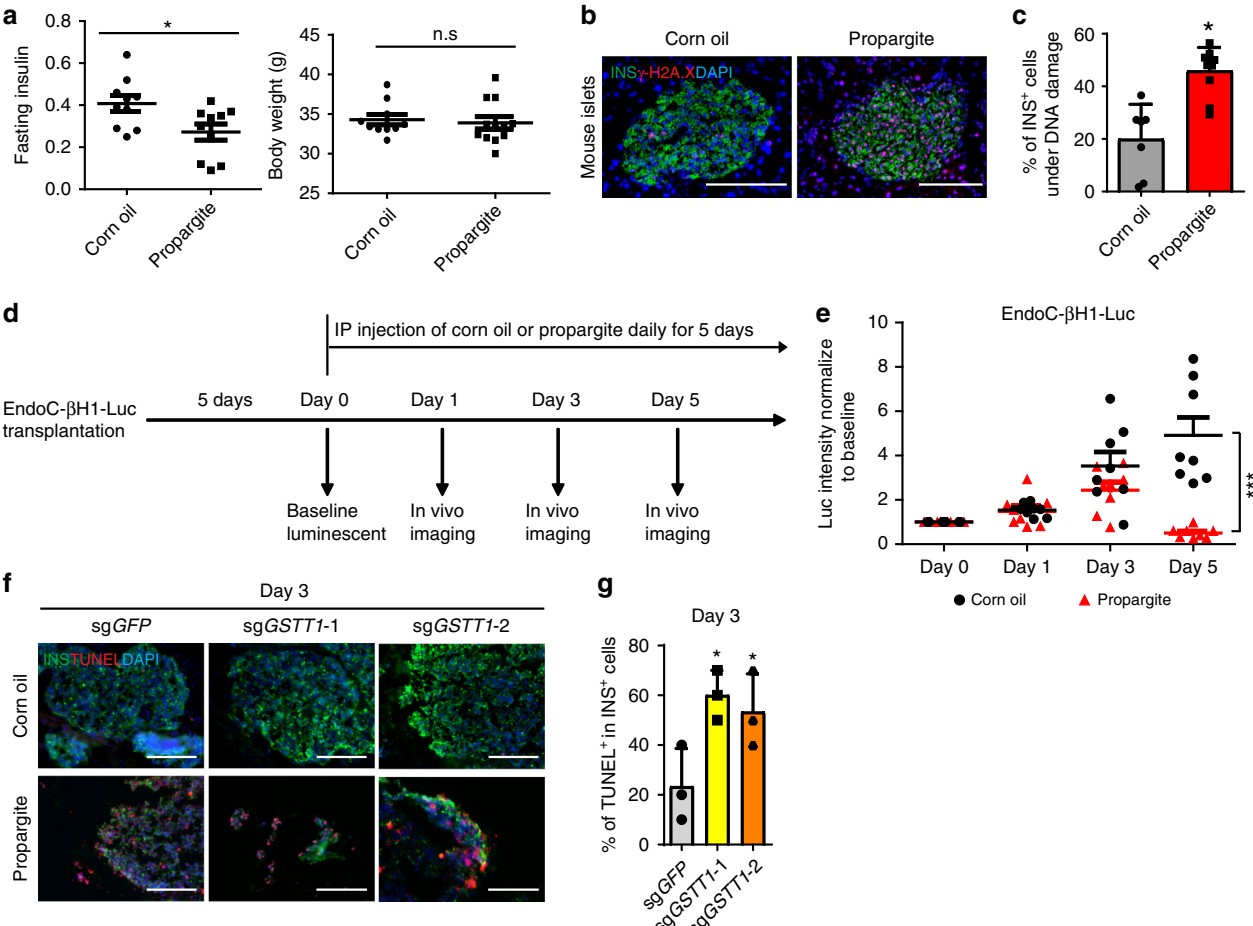

**Fig. 5** Propargite induces β-cell toxicity in vivo. Fasting mouse insulin levels, and body weights in control (corn oil) or 12 mg/kg propargite-treated mice ($n = 10$ mice). $p$ values calculated by one-way repeated measures ANOVA. **b**, **c** Representative images (**b**) and DNA damage rate (**c**) of mouse β-cells in mouse pancreas ($n = 7$ images for corn oil treated mice and $n = 10$ images for Propargite-treated mice). DNA damage rate of mouse β-cells was calculated as the percentage of γ-H2A.X $^+$/INS$^+$ cells in INS$^+$ cells. Scale bars, 100 μm. **d** Scheme of the in vivo imaging analysis. Day 0 is the day the mice are first administrated with corn oil (vehicle) or propargite. **e** Fold-change of luciferase signals from EndoC-βH1-Luc cells transplanted mice administrated with corn oil (vehicle) or propargite during the 5-day course ($n = 8$ mice). $p$ values calculated by one-way repeated measures ANOVA. **f**, **g** Representative images of TUNEL and INS staining (**f**) and Quantification of %TUNEL$^+$ cells (**g**) in EndoC-βH1-Luc cells carrying sg$GSTT1$ or sg$GFP$ on day 3 ($n = 3$ images for each condition). $p$ values calculated by two-way repeated measures ANOVA with a Bonferroni test for multiple comparisons between wt and mutant cells. Related to Supplementary Fig. 6

wells of 96-well plates. A portion of the cells was analyzed by DNA sequencing. For biallelic frameshift mutants, we chose both homozygous mutants or compound heterozygous mutants. Wild-type clonal lines from the corresponding targeting experiments were included as wild-type controls to account for potential non-specific effects associated with the gene targeting process. During differentiation experiments, all lines were re-sequenced to confirm the genotypes. *GSTM1* and *GSTT1* knockout hESC lines were created based on H1 hESCs. The efficiencies for creating biallelic mutant lines were 68% for *GSTM1*$^{-/-}$ cells, 71% for *GSTT1*$^{-/-}$ cells. (Supplementary Table 4).

**Knockout of *GSTT1* in a human β-cell line**. Two sgRNAs targeting two different loci on *GSTT1* gene: sg*GSTT1*-1 and sg*GSTT1*-2 and a sgRNA targeting *GFP* (sg*GFP*)[46] which has no specific targets on the human genome and was used as a control, were cloned into lentiCRISPRv2 vectors (Addgene plasmid # 52961)[47]. EndoC-βH1 cells were infected with the lentivirus carrying Cas9 and sgRNAs. After 2 days of selection with 1 μg/ml puromycin, western blots were used to validate the knockout efficiency.

**RNA sequencing**. The *INS*-GFP$^+$ cells were sorted and RNA from the sorted cells was extracted with Absolutely RNA Nanoprep kit (Agilent Technologies, 400753). The RNA quality was validated with a bioanalyzer (Agilent). The cDNA libraries were synthesized using the TruSeq RNA Sample Preparation kit (Illumina) and sequenced in single-read with the HiSeq2000/1000 sequencer (Illumina) at Weill Cornell Genomics Resources Core Facility. The reads were aligned to the human hg19 reference genome with Tophat2[48]. Gene expression data were analyzed with Cufflinks[49]. To generate heat maps displaying the differential gene expression patterns of different samples, the RPKM values were normalized per gene over all samples. The heat maps were then generated by Heatmap.2 in the R plots package. David functional annotation tool[50] was used for gene ontology analysis and Ingenuity Pathway Analysis tool was used for pathway analysis.

**Immunocytochemistry**. Cells were fixed with 10% (v/v) formalin for 20 min at room temperature (RT) and blocked in a solution of Mg$^{2+}$- and Ca$^{2+}$-free PBS containing 5% horse serum and 0.3% Triton-X for 1 h at room temperature and followed by incubation with primary antibody at 4 °C overnight. The following primary antibodies and dilutions have been used in this study: goat anti-SOX17 (1:500, R&D system, AF1924), rabbit anti-FOXA2 (1:500, Upstate, 07-633), goat anti-PDX1 (1:500, R&D system, AF2419), mouse anti-NKX6.1 (1:100; BCBC, Ab2024), guinea pig anti-INS (1:1000, Dako, A0564), guinea pig anti-GLUCAGON (1:500, Linco, 4031), mouse anti-Ki67 (1:100, Life technologies, 556003), mouse anti-AMYLASE (1:100, Santa Cruz, sc-46657), rabbit anti-Active Caspase-3 (1:500, BD Bioscience, 559565), mouse anti-Troponin T (1:500, Thermo Scientific, MA5-12960), mouse anti-Actin, Cardiac (1:300, Sigma, A9357), rabbit anti-TUJ1 (1:500, Covance, MRB-435P), Chicken anti-MAP2 (1:1000, Abcam, Ab5392), mouse anti-HuC/D (1:500, Invitrogen, A-21271), mouse anti-MAP2 (1:1500, Sigma, M1406), rabbit anti-TH (1:500, Pel freeze, P40101-0), rat-anti-CTIP2 (1:500, Abcam, ab18465), goat anti-FOXA2 (1:200, R&D, 463783). Donkey anti-guinea pig, mouse, goat, rabbit or chicken secondary antibodies conjugated with Alexa-Fluor-488, Alexa-Fluor-594 or Alexa-Fluor-647 fluorophore (1:500, Life technologies) were used. Nuclei were counterstained by DAPI.

**Real-time qPCR**. Total RNA from FACS-sorted *INS*-GFP$^+$ cells was isolated using the Absolutely RNA Nanoprep kit (Agilent Technologies, 400753), quantified with a NanoDrop spectrophotometer (Thermo Scientific), and cDNA was synthesized with a high-capacity cDNA reverse transcription kit (Applied Biosystems, 4374966). Real-time qPCR was performed with a LightCycler 480 (Roche) instrument with LightCycler DNA master SYBR Green I reagents. Statistical significance was determined using a two-tailed Student's *t*-test ($p < 0.005$). Primer sequences are listed in Supplementary Table 1.

**Western blotting analysis**. To extract intracellular protein, cells were placed on ice, washed twice with ice-cold PBS and lysed immediately with RIPA buffer (Life Technologies, 89900) supplemented with a protease inhibitor cocktail (Life Technologies, 78440). Cell lysates were then sonicated and the soluble fraction was collected after centrifugation. Protein amount was quantified by the BCA protein assay (Life Technologies, 23227). To extract extracellular HMGB1, tissue culture media were collected and concentrated using Afyon SDS-PAGE sample preparation kit (advansta, K-02101-025). 20 μg of total protein for each sample was mixed with loading buffer and incubated for 5 min at 95 °C before being loaded into NuPAGE® Novex® 4–12% Tris-Acetate Protein Gels (Life Technologies, NP0321BOX) for electrophoresis. Separated proteins were transferred to a PVDF membrane (Millipore, IPSN07852) after electrophoresis. The membrane was blocked with 5% non-fat milk in TBST (50 mM Tris–HCl at pH 8.0, 150 mM NaCl, 0.1% Tween 20) followed by overnight incubation with the primary antibody at 4 °C. The following antibodies were used with the dilution ratio noted: rabbit anti-HMGB1 (1:1000, Cell Signaling Technology, #6893), rabbit anti-γ-H2A.X (1:1000, Cell Signaling Technology, #9718), rabbit anti-LC3B (1:1000, Cell Signaling Technology, #3868), mouse anti-GSTT1 (1:1000, GeneTex, GTX109250), rabbit anti-GSTM1 (1:1000, R&D system, MAB6894), mouse anti-GAPDH (1:50000,

Abcam, ab8245). A peroxidase conjugated horse anti-mouse IgG secondary antibody (1:20000, Vector Laboratories, PI-2000) or peroxidase conjugated goat anti-Rabbit IgG secondary antibody (1:20000, Vector Laboratories, PI-1000) was used for western blot. ECL signals were developed and detected using the SuperSignal™ West Femto Maximum Sensitivity Substrate (Life Technologies, 34094) and a C-DiGit Chemiluminescent Western Blot Scanner (LI-COR). Uncropped scans of all the blots were shown in Supplementary Fig. 7 in the Supplementary Information.

**Flow cytometry**. Cells were harvested with Accutase, intracellular flow cytometry was performed with Cytofix/Cytoperm™ Fixation/Permeabilization Kit (BD, 554714) according to the user menu. The following primary antibodies and dilutions have been used in this study: C-peptide (1:1000, Millipore, 05-1109), mouse anti-human CD73 (1:1000, BD Pharmingen, 560847) and mouse anti-human CD29 (1:500, eBioscience, 12-0299-41). Cells were stained with primary antibody for 1 h, followed with donkey anti-mouse Alexa-efluor488 secondary antibody (1:500, Life technologies). After washing twice, cells were resuspended in PBS plus 300 nM DAPI (Invitrogen, D21490) and followed by analysis with FACSVantage SE (Becton Dickinson).

**Annexin V cellular apoptosis analysis**. Cells were dissociated by Accutase (Stem Cells) and washed with cold DPBS and then detected with FITC Annexin V Apoptosis Detection Kit I (BD Bioscience, 556547), and analyzed within 1 h by FACSVantage SE (Becton Dickinson).

**Intracellular ROS and total GSH measurement**. Total intracellular ROS were detected using DCFDA-Cellular Reactive Oxygen Species Detection Assay Kit (Abcam, ab113851). Total GSH was measured using GSH-Glo™ Glutathione Assay (Promega, V6911).

**Genotype analysis of *GSTM1* and *GSTT1***. We used a multiplex PCR to detect GSTM1 and GSTT1 as well as an internal control (β-globin) in a one-tube reaction (modified from a published method[51]) to analyze the polymorphism of GSTM1 and GSTM1. Genomic DNA of 3 hESCs and 7 hiPSCs were extracted using the DNeasy Blood & Tissue Kit (QIAGEN, 69506). The 3 genes were amplified from the individual genomic DNAs using primers and a PCR program listed in Supplementary Table 5. The genomic DNA were amplified using the following program: (1) 94 °C, 4 min; (2) 34 cycles of 1 min at 94, 60, and 72 °C; (3) 72 °C for 5 min. The PCR products were electrophoresed on 2% agarose gels stained with ethidium bromide.

**Measurement proparigte's effect on mice**. All animal work has been approved by IACUC committee of Weill Cornell Medical College. CD-1 adult male mice were treated with different doses of propargite or vehicle (corn oil) through intraperitoneal injection for 5 days. Fasting blood glucose level were monitored every day using Abbott FreeStyle Lite blood glucose monitoring system. Then, CD-1 adult male mice were treated with 12 mg/kg propargite or vehicle (corn oil) through intraperitoneal injection for 5 days. Fasting blood was collected via tail nicking following 14–16 h of fasting with ad libitum access to drinking water. Serum mouse insulin levels were assessed by ELISA (80-INSMR-CH01, Alpco). Mice were then killed. The pancreases were embedded in OCT. The tissue was then sectioned at 5 μm for immunohistochemistry staining analysis of DNA damage marker γ-H2A.X and β-cell marker INS.

**Creation of humanized mice and in vivo imaging**. LentiLuc-blast plasmid was constructed based on the LentiCas9-blast vector (Addgene, 52926). The luciferase gene was driven by a constitutively active EFS promoter. EndoC-βH1 cells were infected with the lentivirus followed by 4 days selection with 20 μg/ml blasticidin to create EndoC-βH1-Luc cells. Adult 6–8 week old female NSG immunodeficient mice (Jackson Laboratory) were transplanted with $5 \times 10^5$ EndoC-βH1-Luc cells to create a humanized mouse model. Baseline luminescent levels were obtained 5 days after transplantation. 12 mg/kg propargite or vehicle were then administrated for 5 days via intraperitoneal injection (the starting day referred to as day 0). In vivo imaging was performed on day 1, 3, and 5 with an Xtreme Optical and X-ray small animal imaging system. During the imaging, isoflurane was used for anesthesia. 2 mg/150 μl/25 g body weight luciferin potassium salt (Regis, 360223) was applied to each mouse. To generate EndoC-βH1-Luc cells carrying sg*GSTT1* or sg*GFP*, EndoC-βH1-Luc cells were infected with the lentivirus carrying Cas9 and sgRNAs (sg*GSTT1* or sg*GFP*), and purified with 2 days selection using 1 μg/ml puromycin. Adult 6–8-week-old female NSG-immunodeficient mice (Jackson Laboratory) were transplanted with $5 \times 10^5$ EndoC-βH1-Luc cells carrying sg*GSTT1* or sg*GFP* (control). 12 mg/kg propargite or vehicle were administrated for 3 days via intra-peritoneal injection, mice were then sacrificed at day 3. The xenografts were embedded in OCT. The tissue was then sectioned at 5 μm to detect cell death with TUNEL assays (In Situ Cell Death Detection Kit, TMR red, Sigma, 12156792910) and for immunohistochemistry staining analysis of β-cell marker INS to measure cell death rate in EndoC-βH1-Luc cells carrying sg*GSTT1* or sg*GFP*.

**Statistical analysis**. $n = 3$ independent biological replicates if not otherwise specifically indicated. n.s. indicates non-significant difference. $p$ values were calculated by unpaired two-tailed Student's $t$-test if not otherwise specifically indicated. $*p < 0.05$, $**p < 0.01$, and $***p < 0.001$.

## Data availability

The data that support the findings of this study are available from the corresponding author upon request. RNA-Seq data is available in GEO database (accession number: GSE83699).

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

## Acknowledgements

S.C. is funded by The New York Stem Cell Foundation (R-103), NIH (DP2 DK098093-01, 1 DP3DK111907-01, 1 R01 DK116075-01A1), NCI (U01 CA224326-01), American Diabetes Association (1-17-IBS-019), and Tri-institutional Starr Stem Cell Grant (2014-030). S.C. is New York Stem Cell Foundation-Robertson Investigator. Ti.Z. is funded by a

Family Friendly Postdoctoral (FFPI) program at Weill Cornell. T.W.K is funded by The New York Stem Cell Foundation (NYSCF-D-F50). T.W.K. is a New York Stem Cell Foundation-Druckenmiller Fellow. C.R. is funded by a NIDDK fellowship. This study was also supported by a Shared Facility contract to T.E. and S.C. from the New York State Department of Health (NYSTEM C029156) and by a contract from the New York State Department of Health (NYSTEM C028503) and by R01AG054720 from the National Institute of Aging to L.S. with additional core grant support from P30CA008748. We are also very grateful for technical support and advice provided by Harold S. Ralph in the Cell Screening Core Facility, Jason McCormick in the Flow Cytometry Facility and Lee Cohen-Gould in the Election Microscopy Facility at Weill Cornell Medical College, NY.

## Author contributions

S.C. and Ti.Z. designed the project; Ti.Z. and T.W.K. performed most key experiments; C.C., L.T., S.A., S.M., Z.G., H.Z., M.G., M.C., C.R., R.K., E.A., Z.B., and H.W. performed additional necessary experiments; Tu.Z. and J.Z.X. performed the bioinformatics analysis; Ti.Z., T.W.K., and S.C. analyzed data; Ti.Z., T.W.K., C.R. T.E., L.S., and S.C. wrote the manuscript.

## Additional information

**Competing interests:** The authors declare no competing interests.

