## [Peer Review File · Nature Communications]

Reviewers' comments:

Reviewer #1 (Remarks to the Author):

There is a lot contained in this impressive paper, and I am strongly favorable of accepting the manuscript provided that some serious thought is given to a few issues that are centered on the actual dose causing beta-cell and neuron death, and the "need" for the death-gene signature in RNA-Seq, as described below. For the most part, all of the experiments presented and discussed in the manuscript are done well, and the conclusions are supported by the data. All figures are presented clearly, too.

(1) Although a good part of the manuscript is the testing of various doses of the compounds, I am concerned that even the lowest dose of, for example, propargite, is way above the dose experienced in any human body/organ, even at the estimated ingestion rates that are so nicely quoted in the m/s.

(2) The part on page 7, lines 10-15, seem "obvious" and even simple-minded? The cells died, and pre-empting this event, there was alteration of genes to do with cell death? Could this be altered in presentation style?

(3) It would have been excellent if some of the discovery experimentation here could have been followed up by toxicological experimentation in a whole-animal model, perhaps a rat or mouse. Or, can the authors source other material already published that addresses this issue? Or is one of their points that this chemical is cumulative and very long-lived in the body, potentially accumulating to their test doses? Or becoming more concentrated in specific tissues that are susceptible? Is their citable literature on these issues?

(4) On page 5, the authors write "The chemicals that caused more than 60% reduction in the survival rate of INS+ cells, while affecting less than 20% loss of the INS- cells were picked as primary hits (Fig. 1a and Supplementary Fig. S1b)." I would like the authors to emphasize that this is a very strict gating process, which might miss some important hit compounds.

(5) Page 12, line 13: why is it appropriate to use the word "surprisingly" here - I do not think that it is.

(6) Much of the Discussion section reads disturbingly similar to the abstract and introduction, and this repetitiveness takes away from the flow of the paper. Much of the discussion could be deleted as already covered, or removed from elsewhere. Why couldn't the Discussion section also deal with or raise issues of whole animals vs/ testing cells in vitro, and other issues? How do these tested doses relate to other tox-testing in mice, or point 1 raised above?

[Minor style/text issues, likely to be checked by the copy editor, include the absence of following hyphenation rules in many places. GST is used as an abbreviation on page 8 line 22 before it is first defined much later in the m/s. Also, "interestingly" is used too often - the word should be removed as it is for the reader to determine "interest" based on robustness of data and strength and specificity of findings.]

Reviewer #2 (Remarks to the Author):

This paper uses clonal stem cell derived beta-like cells or dopamine producing neuron-like cells to demonstrate that they are particularly susceptible to the pesticide propargite via DNA damage. The

authors then link this sensitivity to common variants of the GSTT1 gene. In doing so the authors make a strong case for gene-environment interactions that link pesticide exposure to risk for diabetes and Parkinson's. The paper is well written and presented in a linear fashion. There is a large amount of data that are of overall high quality. The methods used are clearly reported and the statistical evaluation of the data appears appropriate. The conclusions are justified by the data that are presented. I have a couple of points that for the authors to consider.

The connection between the propargite-induced cell death in vitro and the epidemiological connection between diabetes/parkinsons and propargite exposure would be stronger if the authors could compare the propargite levels that are (or might reasonably expected to be) present in exposed subjects (e.g. according to ref 41) to the levels that they use to induce death in their in vitro assays. Are the doses used in vitro comparable to what beta cells might be exposed to in exposed humans?

Page 6, line 5/6: Suggesting that the effect of propargite is independent of the maturation state... I think this should read 'dependent' instead. If not I think the statement makes no sense.

Reviewer #3 (Remarks to the Author):

In this manuscript by Zhou et al., the authors describe an hPSC-based platform for identifying toxic interactions between chemicals present in the environment and specific cell types or genotypes. As a proof of concept, they identify propargite as being especially toxic to beta cells and dopaminergic neurons, and elucidate a mechanism by which variants in the GSTT1 locus may predispose cells to propargite toxicity. Overall, this is a significant conceptual advance for the field and the experiments are well-controlled and adequately support most conclusions. I have one concern regarding figure 4, but otherwise congratulate the authors on a nice manuscript.

Comments:

Fig. 4B, C, D: There description of the neuron survival assay is not sufficient to determine if propargite affecting neuron survival or neurogenesis, which may also be occurring simultaneously in these cultures. The authors should describe how they can ensure that propargite is affecting neuronal survival, and if it is necessary, repeat the experiment in a way that this can be definitively shown.

These same concerns also apply to the other cell types tested in Supp Fig. 4.

Discussion: The authors should add text addressing why they think GSTT1 but not GSTM1-null INS+ cells demonstrate vulnerability to propargite despite the fact that both genes have been linked to diabetes by genetic studies.

Thank you for considering our manuscript and for the encouraging initial review. We are very pleased to know that all reviewers appreciated the significance, novelty and data quality of this manuscript. We understand the concerns regarding the experimental details. We carried out additional experiments and analysis to address the reviewers' concerns, which are summarized as 5 figure panels and 6 supplementary figure panels. We hope that these new results, as described in detail below, will alleviate all the reviewers' concerns, and believe that the effort helped generate a much improved study. The reviewers' comments are in black and our responses are in blue.

Reviewer #1 (Remarks to the Author):

There is a lot contained in this impressive paper, and I am strongly favorable of accepting the manuscript provided that some serious thought is given to a few issues that are centered on the actual dose causing beta-cell and neuron death, and the "need" for the death-gene signature in RNA-Seq, as described below. For the most part, all of the experiments presented and discussed in the manuscript are done well, and the conclusions are supported by the data. All figures are presented clearly, too.

(1) Although a good part of the manuscript is the testing of various doses of the compounds, I am concerned that even the lowest dose of, for example, propargite, is way above the dose experienced in any human body/organ, even at the estimated ingestion rates that are so nicely quoted in the m/s.

Response: We did not find any reference describing the dose experienced in human body/organ. Instead, we used both mouse and chimeric mouse models to perform the toxicological experimentation in whole-animal models (Fig. 5), which are detailed below. In summary, 12 mg/kg/day propargite treatment significantly decreases cell numbers and results in increased DNA damage in both mouse and human beta cells. It has been reported that the 8-hour acute absorbed daily dosage (ADD), calculated for aerial applicators handling the wettable powder in water soluble bags, was 5,300 $\mu\text{g}/\text{day}$ per kilogram (kg) of body weight¹. Although not identical, the condition used in mouse experiments is within the range of daily exposure dosage of human.

(2) The part on page 7, lines 10-15, seem "obvious" and even simple-minded? The cells died, and pre-empting this event, there was alteration of genes to do with cell death? Could this be altered in presentation style?

Response: We understand the Reviewer's concern. We have revised the description as below. We are open to further adjusting the text if Reviewer has additional suggestions.

"Gene ontology pathway analysis showed "regulation of DNA-dependent transcription" among the top down-regulated pathways upon propargite treatment (Fig. 2a). In contrast,

genes associated with chromatin assembly and cell death-related processes were up-regulated under those conditions (Fig. 2a). Several genes that were highly up-regulated (fold change >3) in the propargite-treated cells were related to DNA damage, including, DNA damage-inducible transcript 3 (*DDIT3*) and the growth arrest and DNA damage inducible alpha (*GADD45A* or *DDIT1*) (Supplementary Fig. 2a). The upregulation of these genes was further confirmed by qRT-PCR analysis (Fig. 2b).”

(3) It would have been excellent if some of the discovery experimentation here could have been followed up by toxicological experimentation in a whole-animal model, perhaps a rat or mouse. Or, can the authors source other material already published that addresses this issue? Or is one of their points that this chemical is cumulative and very long-lived in the body, potentially accumulating to their test doses? Or becoming more concentrated in specific tissues that are susceptible? Is their citable literature on these issues?

Response: To address reviewer’s concern, we used both mouse and chimeric mouse models (following transplantation of human β -cell line) to perform the toxicological experimentation in whole-animal models (Fig. 5).

First, to investigate if β -cell toxicity is caused by propargite *in vivo*, CD-1 mice were treated with various doses of propargite or corn oil (vehicle) daily for 5 days via intraperitoneal injection. 12 mg/kg propargite treatment slightly increased blood glucose level without significantly affecting body weight (Supplementary Fig. 6a-b). Meanwhile, it led to a significant reduction of insulin levels in the plasma of the fasted mice (Fig. 5a). A significant increase of the percentage of γ -H2AX⁺/INS⁺ in INS⁺ cells of propargite treated mice was also observed, suggesting enhanced rates of propargite-induced DNA damage in mouse pancreatic β -cells *in vivo* (Fig. 5b, 5c).

To investigate the effect of propargite on human pancreatic β -cells *in vivo*, EndoC- β H1-Luc cells were transplanted to NSG immune-deficient mice subcutaneously to create a humanized mouse model (Fig. 5d). The humanized mice were administrated with 12 mg/kg propargite. *In vivo* imaging for luciferase signal showed that luciferase signals from control EndoC- β H1-Luc cells increased gradually during the 5-day course. In contrast, the signal of cells in the propargite administered mice showed a slower rate increase at day 3 and a dramatic decrease at day 5 (Fig. 5e and Supplementary Fig. 6c).

To further confirm that the sensitivity of β -cells to propargite depends on the genetic background, EndoC- β H1-Luc cells carrying sg*GSTT1* were transplanted into immune-deficient mice. EndoC- β H1-Luc cells carrying sg*GFP* were used as a control. At day 3, the survival rate of the EndoC- β H1-Luc cells carrying sg*GSTT1* was associated with higher cell death rate upon propargite treatment as measured with TUNEL assays (Fig. 5f, 5g). Together, the data indicate that *GSTT1*^{-/-} pancreatic β -like cells are hypersensitive to propargite *in vivo*.

It has been reported that the 8-hour acute absorbed daily dosage (ADD) calculated for aerial applicators handling the wettable powder in water soluble bags was 5,300 µg/day per kilogram (kg) of body weight¹. Although not identical, the condition used in mouse experiments is in the range of daily exposure dosage.

(4) On page 5, the authors write "The chemicals that caused more than 60% reduction in the survival rate of INS+ cells, while affecting less than 20% loss of the INS- cells were picked as primary hits (Fig. 1a and Supplementary Fig. S1b)." I would like the authors to emphasize that this is a very strict gating process, which might miss some important hit compounds.

Response: We understand the reviewer's concern about the gating. We agree that the strict gating may miss some hit compounds. We discussed this concern in the discussion section on Page 15 Line 7-11.

(5) Page 12, line 13: why is it appropriate to use the word "surprisingly" here - I do not think that it is.

Response: We understand the reviewer's concern and have removed "surprisingly".

(6) Much of the Discussion section reads disturbingly similar to the abstract and introduction, and this repetitiveness takes away from the flow of the paper. Much of the discussion could be deleted as already covered, or removed from elsewhere. Why couldn't the Discussion section also deal with or raise issues of whole animals vs/ testing cells in vitro, and other issues? How do these tested doses relate to other tox-testing in mice, or point 1 raised above?

Response: To address the reviewer's concern, we have removed some redundant description and added the following sentences in discussion.

"It has been reported that the 8-hour acute absorbed daily dosage (ADD) calculated for aerial applicators handling the wettable powder in water soluble bags was 5,300 µg/day per kilogram (kg) of body weight¹. We used both mouse and humanized mouse models show that daily exposure of propargite causes the decrease of cell number and increase of DNA damage of both mouse and human β-cells."

In addition, we discussed the concern that our screen may miss chemical compounds showing weak acute β-cell toxicity.

[Minor style/text issues, likely to be checked by the copy editor, include the absence of following hyphenation rules in many places. GST is used as an abbreviation on page 8 line 22 before it is first defined much later in the m/s. Also, "interestingly" is used too often - the word should be removed as it is for the reader to determine "interest" based on robustness of data and strength and specificity of findings.]

Response: We have now defined GST in the first appeared on Page 8 Line 21. We have removed "interestingly" at multiple places across the manuscript. Thank you for pointing out this issue.

Reviewer #2 (Remarks to the Author):

This paper uses clonal stem cell derived beta-like cells or dopamine producing neuron-like cells to demonstrate that they are particularly susceptible to the pesticide propargite via DNA damage. The authors then link this sensitivity to common variants of the GSTT1 gene. In doing so the authors make a strong case for gene-environment interactions that link pesticide exposure to risk for diabetes and Parkinson's. The paper is well written and presented in a linear fashion. There is a large amount of data that are of overall high quality. The methods used are clearly reported and the statistical evaluation of the data appears appropriate. The conclusions are justified by the data that are presented. I have a couple of points that for the authors to consider.

Response: We thank the reviewer for appreciating the significance and the novelty of our manuscript.

The connection between the propargite-induced cell death in vitro and the epidemiological connection between diabetes/parkinsons and propargite exposure would be stronger if the authors could compare the propargite levels that are (or might reasonably expected to be) present in exposed subjects (e.g. according to ref 41) to the levels that they use to induce death in their in vitro assays. Are the doses used in vitro comparable to what beta cells might be exposed to in exposed humans?

Response: We did not find any reference describing the dose experienced in human body/organ. To address reviewer's concern, we used both mouse and chimeric mouse models to perform the toxicological experimentation in whole-animal models (Fig. 5), which are detailed as the response to Reviewer 1. In summary, 12 mg/kg/day propargite treatment significantly decreases the cell number and increase DNA damage in both mouse and human beta cells. It has been reported that the 8-hour acute absorbed daily dosage (ADD) calculated for aerial applicators handling the wettable powder in water soluble bags was 5,300 µg/day per kilogram (kg) of body weight¹. Although not identical, the condition used in mouse experiments is in the range of daily exposure dosage of human.

Page 6, line 5/6: Suggesting that the effect of propargite is independent of the maturation state... I think this should read 'dependent' instead. If not I think the statement makes no sense.

Response: We understand concern of the reviewer. We have changed the description to "propargite also causes cell death in glucose-responsive beta-like cells".

Reviewer #3 (Remarks to the Author):

In this manuscript by Zhou et al., the authors describe an hPSC-based platform for identifying toxic interactions between chemicals present in the environment and specific cell types or genotypes. As a proof of concept, they identify propargite as being especially toxic to beta cells and dopaminergic neurons, and elucidate a mechanism by which variants in the GSTT1 locus may predispose cells to propargite toxicity. Overall, this is a significant conceptual advance for the field and the experiments are well-controlled and adequately support most conclusions. I have one concern regarding figure 4, but otherwise congratulate the authors on a nice manuscript.

Response: We thank reviewer for his comments regarding the significance and the novelty of our manuscript.

Comments:

Fig. 4B, C, D: There description of the neuron survival assay is not sufficient to determine if propargite affecting neuron survival or neurogenesis, which may also be occurring simultaneously in these cultures. The authors should describe how they can ensure that propargite is affecting neuronal survival, and if it is necessary, repeat the experiment in a way that this can be definitively shown. These same concerns also apply to the other cell types tested in Supp Fig. 4.

Response: To address reviewer's concern, we performed additional experiments using more highly enriched hPSC-derived cell populations.

First, we re-run the DA differentiation and monitor the purity of day38 DA neurons for our experiments. More than 90% of day38 DA differentiated cells from hPSCs are FOXA2 positive. Among FOXA2 positive cells, more than 95% are MAP2 positive and 75% are TH (a mature DA marker which encodes the rate-limiting enzyme for dopamine synthesis) positive. Less than 1% of the cells are Ki-67 (a marker for cell proliferation) positive (supplemental fig.5a-c). Those data demonstrate that nearly all propargite-treated cells in DA neuron cultures are post-mitotic neurons. Since propargite (1 μ M, 3 μ M) treatment causes a decrease of 40-60% of total cell number, we conclude that it must directly affect DA neuron survival rather than affecting DA neuron differentiation.

In Supplementary Figure 4, we quantified the purity of the various cell types treated with propargite. More than 99% mesenchymal stem cells are CD29+CD73+. BJ fibroblast is a commercially available primary cell line. More than 80 % cells in neuronal population are positively stained by MAP2 antibody. More than 75 % cells in hepatocyte population are positively stained by AIAT antibody (supplemental fig.4a, d, e).

In our previous experiments, the differentiation efficiency to cardiomyocytes was not high. To address this concern, we re-run the cardiac differentiation using a H9-

MYH6:mCherry reporter line and generated cardiomyocytes from the reporter cell line based on the reported protocol², which generates more than 90% of cardiomyocytes. Propargite treatment decreased the number of mCherry+ cardiomyocyte in those highly enriched cultures in a dose-dependent manner (IC50=29.03 μ M, supplemental fig.4b, c, f-h).

Currently, there is no protocol for generating pancreatic exocrine cells at very high efficiency nor protocol for their reliable purification. Accordingly, we decided to take these data out from the revised manuscript.

Overall, we tested the dose-dependent toxicity of propargite across six cell types using highly enriched cell populations, including beta cells, DA neuron, mesenchymal stem cells, BJ fibroblasts, hepatocytes and cardiomyocytes. Propargite treatment typically decreased cell numbers by 40-60% of total cells. Thus, the decrease in cell number is primarily due to cell death, although we cannot completely exclude the possibility that propargite also affects cell differentiation. We discussed this issue on Page 15 Line 11-14.

Discussion: The authors should add text addressing why they think GSTT1 but not GSTM1-null INS+ cells demonstrate vulnerability to propargite despite the fact that both genes have been linked to diabetes by genetic studies.

Response: We have added the following sentences in the discussion section on Page 16 Line 1-7. We are open to further adjust the text if the reviewer has additional suggestions.

“Although both GSTT1-null and GSTM1-null genotype are associated with an increased risk of diabetes³⁻⁵ based on GWAS, only *GSTT1*^{-/-} β -like cells show increased sensitivity to propargite-induced cell death. One possible reason for this finding is that the expression levels of *GSTT1* and *GSTM1* might be different in human versus mouse islets. However, additional experiments will be required to more fully examine the differences between GSTT1-null and GSTM1-null cells.”

Reference:

1. <http://www.cdpr.ca.gov/docs/whs/pdf/hs1527.pdf>.
2. Birket, M.J., *et al.* Expansion and patterning of cardiovascular progenitors derived from human pluripotent stem cells. *Nat Biotechnol* **33**, 970-979 (2015).
3. Zhang, J., Liu, H., Yan, H., Huang, G. & Wang, B. Null genotypes of GSTM1 and GSTT1 contribute to increased risk of diabetes mellitus: a meta-analysis. *Gene* **518**, 405-411 (2013).
4. Saadat, M. Null genotypes of glutathione S-transferase M1 (GSTM1) and T1 (GSTT1) polymorphisms increased susceptibility to type 2 diabetes mellitus, a meta-analysis. *Gene* **532**, 160-162 (2013).
5. Pinheiro, D.S., *et al.* Evaluation of glutathione S-transferase GSTM1 and GSTT1 deletion polymorphisms on type-2 diabetes mellitus risk. *PLoS One* **8**, e76262 (2013).

REVIEWERS' COMMENTS:

Reviewer #1 (Remarks to the Author):

The revised version of this manuscript does read better, and has also included some more experimentation to address the degree of propargite toxicity in beta cells using some in vivo models, such as transplanted human beta-cell lines. I was immediately impressed with the amount of extra work added into the manuscript. There are a few things that should make the m/s details more acceptable and easier to read. There is one remaining big problem for me regarding the claim for a GSTM null state.

(1) It might seem clear to the authors, but the intent of the EndocbetaH1-luc cells is not explicitly written anywhere, I conclude after searching even the methods section. This is a constitutively active (CMV, lentivirus endogenous promoter/enhancer) luciferase, allowing the overall pool size of cells to be estimated in vivo via the amount of light emitted, correct? I do not believe that this basic concept of their new line constructed here for this paper and called EndoCbetaH1luc is explained properly. That is, it's not a reporter for signaling pathway, or mimicking insulin expression.

(2) Some of the figure panels now use very tiny font size and this might lead to problems with the presentation of the figures. They look fine when they are expanded a lot on my computer monitor. Maybe this point is irrelevant for online viewing.

(3) Figure 3d – what is the remaining signal in both of the null-state cell samples 1 and 2? Authors state: "Each indel mutation created an early frame shift that generated null alleles as confirmed by western blotting experiments in mutant hESC-derived cells (Fig. 3d)." How does presence of a band at the same size as the WT show that these are null alleles? It does not, to me. Are these lanes mistakenly from +/- cell lines by any chance?

(4) Figure 3 b and c – the symbols and lines are all presented directly within the exact same vertical line, making them overlap and just too difficult to examine, and they should be spread laterally within the two different cell state columns.

Reviewer #3 (Remarks to the Author):

The authors have experimentally addressed my concerns on Fig. 4 and Supplementary Fig. 4 by adding quantification or alternative differentiation protocols that more clearly show that the reductions of cell numbers are due in large part to cell death. I believe this manuscript is now suitable for publication.

Point-to-Point Response.

Reviewer #1 (Remarks to the Author):

The revised version of this manuscript does read better, and has also included some more experimentation to address the degree of propargite toxicity in beta cells using some *in vivo* models, such as transplanted human beta-cell lines. I was immediately impressed with the amount of extra work added into the manuscript. There are a few things that should make the m/s details more acceptable and easier to read. There is one remaining big problem for me regarding the claim for a GSTM null state.

(1) It might seem clear to the authors, but the intent of the EndocbetaH1-luc cells is not explicitly written anywhere, I conclude after searching even the methods section. This is a constitutively active (CMV, lentivirus endogenous promoter/enhancer) luciferase, allowing the overall pool size of cells to be estimated *in vivo* via the amount of light emitted, correct? I do not believe that this basic concept of their new line constructed here for this paper and called EndoCbetaH1luc is explained properly. That is, it's not a reporter for signaling pathway, or mimicking insulin expression.

Response: We understand the reviewer's concern. The reviewer is correct that the luciferase gene is driven by a constitutively active EFS promoter. After infection and selection, all remaining EndoC- β H1 cells express luciferase. This line is used for *in vivo* imaging to monitor the survival of the EndoC- β H1 cells.

To avoid confusion, we revised the description of the EndoC- β H1-Luc cells as below on Page 14 Line 1.

“To investigate the effect of propargite on human pancreatic β -cells *in vivo*, we infected EndoC- β H1 cells with a lenti-virus carrying constitutively expressed luciferase to generate the luciferase-expressing EndoC- β H1 cells (EndoC- β H1-Luc cells). The EndoC- β H1-Luc cells were then transplanted into NSG immune-deficient mice to create a humanized mouse model (Fig. 5d).”

We also updated the method section at Page 37 Line 5.

(2) Some of the figure panels now use very tiny font size and this might lead to problems with the presentation of the figures. They look fine when they are expanded a lot on my computer monitor. Maybe this point is irrelevant for online viewing.

Response: Thank you for the suggestion. We have increased the font size in all the figures.

(3) Figure 3d – what is the remaining signal in both of the null-state cell samples 1 and 2? Authors state: “Each indel mutation created an early frame shift that generated null alleles as confirmed by western blotting experiments in mutant hESC-derived cells (Fig. 3d).” How does presence of a band at the same size as the WT show that these are null alleles? It does not, to me. Are these lanes mistakenly from +/- cell lines by any chance?

Response: We understand the reviewer's concern. The two selected GSTM1 KO clones were both compound heterozygous mutants: *GSTM1*^{-/-} -1 (M109fs/M109fs) and *GSTM1*^{-/-} -2 (M109fs/M105fs) (Supplementary Figure 3e).

We re-run the western blot using freshly prepared samples. Our new data confirm that there is minimal signal in *GSTM1*^{-/-} -1 and *GSTM1*^{-/-} -2 cells. The GSTM1 (25KD) signal is likely caused by background binding of the GSTM1 antibody. We have replaced the figure 3d with the new blots.

(4) Figure 3 b and c – the symbols and lines are all presented directly within the exact same vertical line, making them overlap and just too difficult to examine, and they should be spread laterally within the two different cell state columns.

Response: Thank you for the suggestion. We re-created Figure 3b and c to address this point.